# Comparing different versions of computer-aided detection products when reading chest X-rays for tuberculosis

**Zhi Zhen Qin**[1]*, **Rachael Barrett**[1], **Shahriar Ahmed**[2], **Mohammad Shahnewaz Sarker**[2], **Kishor Paul**[2], **Ahammad Shafiq Sikder Adel**[2], **Sayera Banu**[2‡], **Jacob Creswell**[1‡]

**1** Stop TB Partnership, Le Grand-Saconnex, Geneva, Switzerland, **2** International Centre for Diarrhoeal Disease Research, Bangladesh (icddr,b), Dhaka, Bangladesh

‡ These authors are joint senior authors on this work.
* zhizhenq@stoptb.org

**Data Availability Statement:** All numeric data and codes used in this manuscript are available here: https://github.com/ZZQin/MachineBGD/tree/master/2.0%20Version%20Comparison.

## Abstract

Computer-aided detection (CAD) was recently recommended by the WHO for TB screening and triage based on several evaluations, but unlike traditional diagnostic tests, software versions are updated frequently and require constant evaluation. Since then, newer versions of two of the evaluated products have already been released. We used a case control sample of 12,890 chest X-rays to compare performance and model the programmatic effect of upgrading to newer versions of CAD4TB and qXR. We compared the area under the receiver operating characteristic curve (AUC), overall, and with data stratified by age, TB history, gender, and patient source. All versions were compared against radiologist readings and WHO's Target Product Profile (TPP) for a TB triage test. Both newer versions significantly outperformed their predecessors in terms of AUC: CAD4TB version 6 (0.823 [0.816–0.830]), version 7 (0.903 [0.897–0.908]) and qXR version 2 (0.872 [0.866–0.878]), version 3 (0.906 [0.901–0.911]). Newer versions met WHO TPP values, older versions did not. All products equalled or surpassed the human radiologist performance with improvements in triage ability in newer versions. Humans and CAD performed worse in older age groups and among those with TB history. New versions of CAD outperform their predecessors. Prior to implementation CAD should be evaluated using local data because underlying neural networks can differ significantly. An independent rapid evaluation centre is necessitated to provide implementers with performance data on new versions of CAD products as they are developed.

## Author summary

The World Health Organization recommended the use of artificial intelligence (AI)-powered computer-aided detection (CAD) for TB screening and triage in 2021. One year on, we comprehensively compare the performance of the newest versions of two CAD (CAD4TB and qXR) to their WHO-evaluated predecessors. We found that both newer

**Funding:** This project was funded by Global Affairs Canada through the Stop TB Partnership's TB REACH Initiative (grant number STBP/TBREACH/GSA/W5-24). The funders had no role in study design, data collection and analysis, decision to publish, or preparation of the manuscript.

**Competing interests:** The authors have declared that no competing interests exist.

versions significantly improved upon their predecessor's ability to detect TB, performing better than the human readers. We also showed that the AI underlying new software versions can differ remarkably from the old and resemble an entirely new product altogether. We further demonstrate that, unlike laboratory diagnostic tools, CAD software updates could significantly impact the selection of appropriate threshold scores, the number of people with TB detected and cost-effectiveness. With newer CAD versions being rolled out almost annually, our results therefore underscore the need for rapid evidence generation to evaluate newer CAD versions in the fast-growing medical AI industry.

## Introduction

Several computer-aided detection (CAD) products for TB have emerged and can provide an automated and standardized interpretation of digital chest X-ray (CXR) based on artificial intelligence.[1] Recent evaluations of CAD's ability to detect TB-related abnormalities report performance comparable to (or better than) human readers.[2] In March 2021, the World Health Organization (WHO) reviewed impartial evaluations of three CAD products, and made the landmark decision to update international TB screening policy to include the use of CAD on CXR of individuals ≥15 years.[3] Under the WHO guidance, other CAD products may be utilized providing their performance matches those reviewed in the guideline.

The emergence of CAD as a high-performing tool for screening and triage has been a different exercise compared to new lab diagnostics, with newer software versions being available rapidly. The speed of this progress presents a challenge to the relevance of current CAD literature and the policy it informs. Two of the products reviewed during the WHO guideline development process published in 2021, CAD4TB V6 (Delft Imaging Systems, the Netherlands) and qXR V2 (Qure.ai, India), have already been updated. Further, modern CAD software is developed using the AI technique that works by mimicking human brain–neural networks.[1] However, the inner workings of commercial CAD software are challenging to understand for both general audiences and developers, because the nature of neural networks are akin to a black box and the underlying algorithms are the utmost business secret. Therefore, for medical professionals, who will not know any real commercial AI software's inner workings, confidence in the ability of CAD software to detect TB should be earned by comprehensive and unbiased software evaluations that measure different performance indicators on real-world datasets. Only one study, also outdated by new software versions, assessed and compared consecutive versions of a single CAD product.[4] More broadly, there is a lack of research in quantifying differences in the programmatic impact between software versions to advice users and TB programmes what and if adjustment is needed when using a software tools that update on a yearly or more rapid basis. We therefore compare the performance of two WHO-evaluated CAD product versions with the subsequent versions, using bacteriological evidence as the reference standard.

## Materials and methods

The study evaluated CAD4TB versions 6 and 7, and qXR versions 2 and 3.[5] Both CAD products read CXR images and calculate an abnormality score representing the likelihood that TB-associated abnormalities are present in an image. A dichotomous result (TB-associated abnormalities present or absent) is arrived at by setting a threshold abnormality score, above which the algorithm suggests that TB-associated abnormalities are present, and that the individual should undergo further confirmatory testing.[1] The outputs also include heat maps indicating

the location of abnormalities. Both products had been reviewed by the WHO Guidelines Development Group and approved for use in TB triage and screening (in individuals ≥15 years) in 2021.[3] The dataset used in this evaluation is taken from the Stop TB Partnership's TB REACH CXR Evaluation Centre.[5]

## CXR sample collection

Every individual ≥15 years old visiting one of three TB screening centres set up by icddr,b in Dhaka, Bangladesh were verbally screened for TB symptoms–cough, shortness of breath, weight loss, haemoptysis–and received a CXR. The image was then read by one of three radiologists registered with the Bangladesh Medical and Dental Council. The radiologists were blinded to any information except age and sex. They classified each image as 'normal' or 'abnormal' (including any abnormality, whether consistent with TB or not).[6] Regardless of the CXR results, all individuals were asked to submit a fresh spot sputum sample for testing with Xpert MTB/RIF (Xpert) assay. Xpert provided a bacteriological reference standard, confirming the presence (Bac+) or absence (Bac-) of mycobacterium tuberculosis.

For this study, this dataset was sampled using case control sampling, with a 2 to 1 match of 8,582 Bac- and 4,308 Bac+ CXR according to the reference standard, resulting in a dataset of 12,890 CXRs which were read by all four software versions.

CAD reading was performed retrospectively during sessions where CAD4TB and qXR were installed on Stop TB Partnership's Secure File Transfer Protocol server storing the de-identified CXR images. CAD developers were not granted access to the evaluation dataset before or after the reading, and reading was performed blind of all clinical and demographic information, and without any prior AI training. Unique identifiers were used to group server datasets for analytical purposes. Only co-authors had access to the dataset.

## Data analysis

To compare the accuracy of newer against older versions, receiver operating characteristic (ROC) curves were plotted and the area under the ROC curves (AUC) was calculated as a general indication of product version accuracy over the entire abnormality score range.

A paired one-sided t-test was performed to test whether the average CAD4TB v7 score is less than the average CAD4TB v6 score. The same was performed to test if the average qXR v2 score was less than the average qXR v3 score. We also constructed histograms of the abnormality scores of the different software versions disaggregated by bacteriological status. To examine how performance changes across threshold scores we evaluated the cost-saving of each product version in a hypothetical triage situation with CXR from 20,000 adults would be interpreted by each CAD version and only those with an abnormality score above a threshold value would receive an Xpert diagnostic test. We assumed the prevalence of Bac+ TB in the population was 19%, as in the principal study, then calculated the sensitivity of each version and number of Xpert assays hypothetically needed.[2,7]

To compare human with AI performance, we calculated the sensitivity and specificity of the Bangladeshi radiologists and the threshold score each version would need to match this sensitivity. We then compared the difference in specificity between human readers and each CAD version using the McNemar test for paired proportions.

We also compared version performance at target sensitivity and specificity values according to the WHO's target product profile (TPP) for a TB triage test of sensitivity ≥90% and specificity ≥70%.[8] Similarly, the threshold of each version was chosen to match the sensitivity target value, and likewise for specificity targets. Finally, subgroup analysis was performed by stratifying AUCs by gender, patient source, age group, and history of TB. For the same subgroups, we

also calculated human reader sensitivity and specificity. All calculations were done using the statistical software R, v 3.6.0 (R Computing, Vienna, Austria).

### Ethics

All enrolled participants provided informed written consent, those under 18 years of age gave assent in addition to parent's or guardian's consent, their medical data were anonymized, and ethical approval was obtained approval from the Research Review Committee and the Ethical Review Committee at icddr,b.

### Role of CAD developers

AI developers had no role in study design, data collection, analysis plan, or writing of the publication.

## Results

The median age of the 12,890 participants was 42.0 [29.0, 57.0]; fewer than one third (32.0%) were female; and 1,991 individuals (15.5%) had a history of TB (Table 1). All individuals reported TB-related symptoms, the most common being a cough, reported by 11,651 individuals (90.5%), followed by fever (10,323; 80.2%), weight loss (8,440; 65.6%), shortness of breath (6,742; 52.4%) and haemoptysis (1,625; 12.6%). 777 (18.0%) had a high bacterial burden and rifampicin resistance was detected in 206 (1.6%) people. Most were referred from private (9,538 [75.9%]), public (231 [1.8%]), or DOTS (1,284 [10.2%]) facilities. 1,341 (10.7%) were walk-ins, while 86 (0.7%) came from community screening and 82 (0.7%) were contacts.

4,308 individuals (33.4%) were Bac+ according to the reference standard. Radiologists graded 4,147 (32.2%) of all CXRs as normal, but only 5% of the CXRs of Bac+ individuals as normal; and 3,932 (45.8%) of the CXRs of Bac- individuals as normal.

The median score allocated to Bac+ individuals by CAD4TB increased from 81.0 to 97.2 between version 6 and 7, while the median score for Bac- individuals decreased dramatically from 53.0 to 9.6. qXR version 2 attributed slightly higher scores to Bac+ individuals than v3, with a median of 91.5 compared to 89.0. qXR v3 allocated lower scores to Bac- people (median = 12.0) than v2. Ten percent of Bac+ individuals had a CAD4TB v6 score less than 61, a CAD4TB v7 score less than 49.8, a *qXR v2* score less than 61.7, and a qXR v3 score less than 59. Ten percent of Bac- individuals had a CAD4TB v6 score greater than 84, a CAD4TB v7 score greater than 89.9, a qXR v2 score greater than 89.9, and a qXR v3 score greater than 84.

### Overall performance and modelled programmatic impacts

CAD4TB v7 had a significantly higher AUC than v6, 0.903 (95% CI: 0.897–0.908) compared to 0.823 (0.816–0.830). qXR version 3 significantly outperformed v2, with AUCs of 0.906 (95% CI: 0.901–0.911) and 0.872 (0.866–0.878) respectively (S1 Fig Right, S1 Table). The improvement in CAD4TB was greater, but qXR v2 was significantly better than CAD4TB V6 at baseline.

(S1 Fig Left) Receiver Operating Characteristic (ROC) Curves of CAD4TB v6 and v7. (S1 Fig Right) ROC curve of qXR v2 and v3. (S2A Fig) Sensitivity versus threshold abnormality score for qXRv2 and v3. (S2B Fig) Xpert tests saved versus threshold abnormality score for qXRv2 and v3. (S2C Fig) Sensitivity versus threshold abnormality score for CAD4TBv6 and v7. (S2D Fig); Xpert tests saved versus threshold abnormality score for CAD4TBv6 and v7.

The sensitivity of qXR versions 2 and 3 was similar across different threshold scores. When a threshold abnormality score was between 0 and 75, the sensitivity of both qXR versions was

**Table 1. Characteristics of the 12,890 participants.**

| | Overall (%) | Xpert Positive (%) | Xpert Negative (%) | p test |
|---|---|---|---|---|
| n | 12890 | 4308 | 8582 | |
| Age (median [IQR]) | 41.0 [29.0, 57.0] | 37.0 [27.0, 53.0] | 43.0 [31.0, 58.0] | <0.001 |
| Age group | | | | <0.001 |
| 15< 25 years | 1,553 (12.0) | 774 (18.0) | 779 (9.1) | |
| 25< 60 years | 8,606 (66.8) | 2,786 (64.7) | 5,820 (67.8) | |
| 60 years+ | 2,731 (21.2) | 748 (17.4) | 1,983 (23.1) | |
| Gender = F/M (%) | 4,127/8,763 (32.0/68.0) | 1,242/3,066 (28.8/71.2) | 2,885/5,697 (33.6/66.4) | <0.001 |
| Any Symptoms (%) | 12,890 (100.0) | 4,308 (100.0) | 8,582 (100.0) | |
| Cough (%) | 11,651 (90.5) | 4,012 (93.2) | 7,639 (89.2) | <0.001 |
| Fever (%) | 10,323 (80.2) | 3,745 (87.0) | 6,578 (76.7) | <0.001 |
| Shortness of Breath (%) | 6,742 (52.4) | 2,317 (53.9) | 4,425 (51.7) | 0.019 |
| Weight Loss (%) | 8,440 (65.6) | 3,267 (75.9) | 5,173 (60.4) | <0.001 |
| Haemoptysis (%) | 1,625 (12.6) | 547 (12.7) | 1,078 (12.6) | 0.85 |
| TB History (%) | 1,991 (15.5) | 715 (16.6) | 1,276 (14.9) | 0.011 |
| Xpert positive (Bac+) (%) | 4,308 (33.4) | 4,308 (100.0) | 0 (0.0) | <0.001 |
| Bacterial Burden (%) | | | | 0.485 |
| High | 777 (18.0) | 776 (18.0) | 1 (50.0) | |
| Medium | 1,497 (34.7) | 1,497 (34.8) | 0 (0.0) | |
| Low | 1,289 (29.9) | 1,288 (29.9) | 1 (50.0) | |
| Very Low | 745 (17.3) | 745 (17.3) | 0 (0.0) | |
| RIF Result (%) | | | | <0.001 |
| Detected | 206 (1.6) | 206 (4.8) | 0 (0.0) | |
| Indeterminate | 21 (0.2) | 21 (0.5) | 0 (0.0) | |
| Not Detected | 12,662 (98.2) | 4,080 (94.7) | 8,582 (100.0) | |
| Patient Source | | | | <0.001 |
| Community screening | 86 (0.7) | 16 (0.4) | 70 (0.8) | |
| Contacts | 82 (0.7) | 9 (0.2) | 73 (0.9) | |
| Private Referral | 9,538 (75.9) | 3,382 (80.5) | 6,156 (73.6) | |
| DOTS Retesting | 1,284 (10.2) | 471 (11.2) | 813 (9.7) | |
| Public Referral | 231 (1.8) | 102 (2.4) | 129 (1.5) | |
| Walk-in | 1,341 (10.7) | 219 (5.2) | 1,122 (13.4) | |
| Radiology Result (%) | | | | <0.001 |
| X-Ray Abnormal | 8,743 (67.8) | 4,093 (95) | 4,650 (54.2) | <0.001 |
| X-Ray Normal | 4,147 (32.2) | 215 (5.0) | 3,932 (45.8) | |
| CAD median [10%, 90% percentile] | | | | |
| CAD4TBv6 | 66.0 [33, 91 0] | 81.0 [61.0, 95.0] | 53.0 [25.0, 84.0] | <0.001 |
| CAD4TBv7 | 45.6 [1.0, 99.1] | 97.2 [49.8, 99.6] | 9.6 [0.6, 89.8] | <0.001 |
| qXRv2 | 65.6 [7.6, 94.4] | 91.5 [61.7, 96.0] | 28.8 [6.47, 89.9] | <0.001 |
| qXRv3 | 53.0 [2, 93] | 89.0 [59.0, 95.0] | 12.0 [2.0, 84.0] | <0.001 |

high, displaying a similar gradual reduction as threshold increases (S2A Fig, S2 Table). In the modelling population, both qXR versions begin saving large numbers of diagnostic tests initially, (S2B Fig). For example, at a threshold score of 50, there is no significant difference in the sensitivities of qXR versions 2 and 3 (93.0% [95% CI: 92.2–93.8%] and 92.5% [91.7–93.3], respectively). V3, however, was more specific and as a result saved 57.8% of diagnostic tests compared to 50.9% saved by v2.

Overall CAD4TB versions 6 and 7 show a vastly altered relationship between abnormality score, sensitivity, and number of Xpert tests saved (S2C and S2D Fig). At a low threshold score (until approximately 48), CAD4TB v6 remains close to 100% sensitive while v7 maintains high sensitivity (80–100%) over most of its threshold score range, only at a threshold of 81 or higher falling below 80%. Similarly, S2D Fig indicates that versions 6 and 7 offer vastly different cost savings and scores needed to achieve them. Until thresholds of approximately 45 are reached, fewer than 20% of diagnostic tests are saved by v6 because of the linear initial relationship between abnormality score and diagnostic test saving. In contrast, v7 results in a steep initial increase. Until a threshold of 75, greater numbers of Xpert tests can be saved using CAD4TB v7.

## Abnormality score distributions

The average CAD4TB v7 score was significantly less than the average CAD4TB v6 score with a mean difference of -15.0 (p-value < 2.2e-16). The average qXR v3 score was also significantly less than the average qXR v2 score with a mean difference of -7.7 (p-value < 2.2e-16).

The histogram of the abnormality scores for Bac+ and Bac- individuals show clear overlap for CAD4TB v6, but none for v7 (S3 Fig), indicating that CAD4TB v7 could differentiate most Bac+ from Bac- individuals at a score between 80 and 85. Similar observations were noted for qXR, the newer version providing improved separation of Bac+ and Bac- individuals, with fewer false negative and false positive cases.

Bac. Pos.–individuals with TB according to the Xpert reference standard. Bac. Neg.–individuals without TB according to the Xpert reference standard.

All versions allocated higher scores to Bac- people with a history of TB than to those without. However, large numbers of outliers remain–many Bac+ individuals have extremely low CAD4TB v7 scores or qXR v3 scores.

## Comparison against human readers

The human radiologist's sensitivity was 88.2% (87.2–89.1%) and specificity was 62.8%, (61.8–63.9%) (Table 2). Matching the sensitivity, all versions had significantly greater specificity than human radiologists except for CAD4TB v6, which was similar. CAD4TB v7 significantly improved on its predecessor's specificity, outperforming human radiologists with specificity of 76.0% (75.1–76.9%), compared to 62.8% (61.8–63.9%) by human readers, and 64.1% (63.1–65.2%) of CAD4TB v6.

While version 2 of qXR was significantly more specific than human radiologists, V3 improved more and was 13.7% (12–15%) more specific than Bangladeshi radiologists while matching sensitivity.

## Comparison against WHO TPP

The earlier versions of both products did not meet the WHO TPP, whereas the newly released version exceeded the target.

At 90% sensitivity, the newer versions of CAD4TB and qXR significantly improved upon their predecessors and obtained specificities of 72.8% (95% CI: 71.9–73.8%) and 74.2% (73.3–75.1%) for qXR, respectively (Table 3). For CAD4TB v7 a lower threshold score yielded 90% sensitivity compared to v6, while no such change was observed between qXR versions.

Although qXR v2 came close, only the updated versions met the 90% sensitivity target at 70% specificity. CAD4TB v7 had sensitivity of 91.5% (90.6–92.3%) compared to 80.9% (79.7–82.1%) of v6; qXR v3 met the TPP with sensitivity of 92.3% (91.5–93.1%), while v2 came close at 88.2% (87.2–89.2%) (Table 3).

**Table 2. Comparison of product versions against human radiologists.**

| | Threshold abnormality score | Sensitivity (95% CI) | Specificity (95% CI) | Absolute difference in specificity between CAD and radiologists (95% CI) |
|---|---|---|---|---|
| Radiologists' performance | | 88.2%, (87.2–89.1%) | 62.8%, (61.8–63.9%) | - |
| CAD4TB (version) performance | | | | |
| CAD4TB (v6) | 64 | 88.3% (87.3–89.2%) | 64.1% (63.1–65.2%) | 1.32% (0–3%) |
| CAD4TB (v7) | 58 | 88.2% (87.2–89.2%) | 76.0% (75.1–76.9%) | 13.2% (12–15%) |
| qXR (version) performance | | | | |
| qXR (v2) | 67 | 88.2% (87.2–89.2%) | 70.2% (69.2–71.2%) | 7.38% (6–9%) |
| qXR (v3) | 65 | 88.4% (87.4–89.3%) | 76.6% (75.6–77.4%) | 13.7% (12–15%) |

## Subgroup analysis

The AUCs of both newer CAD products are higher than the previous versions across all subgroups. Overall, the AUCs of all CAD product versions were significantly higher in new cases compared to people with a history of TB: ranging from 0.846–0.918 for new cases and 0.706–0.841 for those who had TB previously. Despite comparable sensitivity, human readers also performed worse in this group with specificity of 37.62% (34.95–40.34%) compared to 67.2%, (66.1–68.3%) where there was no TB history. (Table 4). All product versions also performed significantly worse in older populations, as did human readers. No significant gender difference was noted for CAD, though human readers were less specific in males than females.

Newer product versions were more proficient at accurately classifying CXRs from people with a history of TB and older individuals, especially CAD4TB v7 compared to v6 (S4 Fig, S3 Table). V3 of qXR performed significantly better than its predecessor in older and middle-aged groups, while v7 of CAD4TB significantly outperformed its predecessor in all age groups and was the only algorithm not to perform worse in middle-aged than in younger age groups.

Patient source was a conspicuous factor. All versions performed significantly better among walk-ins than DOTS-retested and private referrals and human readers showed the same bias. qXR versions 2 and 3, and CAD4TB v6 also performed worse in private referrals in general, but this shortcoming was not carried forward into CAD4TB v7. Human reader specificity was

**Table 3. The performance of old and new versions of CAD4TB and qXR when fixed to 90% sensitivity and to 70% specificity.**

| CAD software & version | Threshold abnormality score | Sensitivity (95% CI) | Specificity (95% CI) |
|---|---|---|---|
| CAD4TB | | | |
| Fixing sensitivity at 90% | | | |
| v6 | 62 | 89.9% (88.9–90.8%) | 61.4% (60.3–62.4%) |
| v7 | 50 | 89.9% (89.0–90.8%) | 72.8% (71.9–73.8%) |
| Fixing specificity at 70% | | | |
| v6 | 69 | 80.9% (79.7–82.1%) | 70.6% (69.7–71.6%) |
| v7 | 44 | 91.5% (90.6–92.3%) | 69.9% (68.9–70.9%) |
| qXR | | | |
| Fixing sensitivity at 90% | | | |
| v2 | 61 | 90.3% (89.4–91.2%) | 66.8% (65.8–67.8%) |
| v3 | 60 | 90.0% (89.0–90.9%) | 74.2% (73.3–75.1%) |
| Fixing specificity at 70% | | | |
| v2 | 67 | 88.2% (87.2–89.2%) | 70.2% (69.2–71.2%) |
| v3 | 51 | 92.3% (91.5–93.1%) | 70.0% (69.0–71.0%) |

**Table 4. The sensitivity and specificity of human readers in these subgroups.**

| Subgroup | Human Reader Sensitivity | Human Reader Specificity |
|---|---|---|
| Overall | 88.2%, (87.2–89.1%) | 62.8%, (61.8–63.9%) |
| Age Group | | |
| Young Age | 89.4%, (87.0–91.5%) | 75.6%, (72.4–78.6%) |
| Middle Age | 88.8%, (87.6–90.0%) | 66.2%, (65.0–67.4%) |
| Old Age | 84.5%, (81.7–87.0%) | 47.9%, (45.7–50.1%) |
| TB History | | |
| Without TB history | 88.2%, (87.1–89.2%) | 67.2%, (66.1–68.3%) |
| With TB history | 88.4%, (85.8–90.6%) | 37.6%, (35.0–40.3%) |
| Gender | | |
| Female | 87.4%, (85.5–89.2%) | 66.7%, (64.9–68.4%) |
| Male | 88.5%, (87.3–89.6%) | 60.9%, (59.6–62.1%) |
| Patient Source | | |
| Private Referral | 87.9%, (86.8–89.0%) | 59.1%, (57.8–60.3%) |
| Public Referral | 86.3%, (78.0–92.3%) | 58.1%, (49.1–66.8%) |
| DOTS Retesting | 92.4%, (89.6–94.6%) | 60.3%, (56.8–63.7%) |
| Walk-in | 85.8%, (80.5–90.2%) | 82.0%, (79.6–84.2%) |
| Community screening | 93.8%, (69.8–99.8%) | 72.9%, (60.9–82.8%) |
| Contacts | 100.0%, (66.4–100.0%) | 76.7%, (65.4–85.8%) |

also slightly lower in this group- 59.1% (57.8–60.3%) compared to 62.8% (61.8–63.9%) overall. Similarly, all products and human readers performed significantly worse among public referrals compared to walk-ins except CAD4TB v7. Between other patient sources, no significant differences were observed in CAD, where human readers also displayed slightly higher specificity in community screening (72.9% [60.9–82.8%]) than private referrals (59.1% [57.8–60.3%]).

## Discussion

This is the first study that compares the newer versions of the WHO-reviewed CAD products, qXR and CAD4TB. Both new software versions exceeded the performance of their WHO-evaluated previous versions and met the TPP targets. Our findings illustrate measurable improvements achieved by new versions of software. However, the opacity of the technology makes it difficult to predict how these changes will impact programmes since new versions of products can involve significant changes in the underlying neural network and should therefore be evaluated as if they were new products altogether to verify their performance maintains the level of those in the WHO guideline update.

A given threshold score deployed with different versions of the same CAD product will not always be associated with the same sensitivity and Xpert saving, as exemplified by CAD4TB v7 compared to v6. The improvement seen with v7 may be attributed to a large difference in the underlying neural network, demonstrated by the box plots of the abnormality scores of the two versions. In contrast, the two versions of qXR showed more nuanced improvement and the underlying classification algorithm remains largely similar between versions, although the newer can save more confirmatory tests while keeping the sensitivity the same. For example, using 60 as the threshold score with CAD4TB v6 achieved 92% sensitivity and saved about 43% of Xpert tests. If the software was then updated to v7 and the same threshold used, sensitivity would reduce to 88% and the programme would now save 55% of diagnostic tests. New

software updates will likely necessitate the adjustment of the threshold score to maintain performance analogous to that of the previous version.

In general, both the older and newer versions of qXR and CAD4TB outperformed human readers, except CAD4TB v6 which performed similarly. These findings are in line with previous research.[9,10] The improvement in performance we observed in CAD4TB agrees with a previous study describing improvement in version 6 compared to predecessors.[4]

However, algorithms can be further refined to improve performance for subgroups such as older age groups and those with a history of TB.[2] Current weaknesses suggest a flaw in current training practices that may be limiting CAD accuracy, even in newer versions. However, human reader bias mirrored that of CAD when it came to older age groups and those with a history of TB. As new versions are automatically rolled out to users globally, their programmatic implications should be routinely monitored to ensure they serve all populations in need. A rapid evaluation centre, with access to diverse datasets from different regions of the world, will be key to meeting this need.

This study has a few limitations. Firstly, owing to logistic and budgetary constraints, we did not use culture as the reference standard, meaning that some people with Xpert-negative, culture-positive TB might have been incorrectly labelled as not having the disease. We also did not have access in Bangladesh to Xpert Ultra, which is more sensitive than Xpert. Due to the small number of asymptomatic individuals by symptoms or test for HIV, subgroup analysis was not performed on these groups. The study population also excludes children under 15 due to protocol limitation.

## Conclusion

Updated versions of CAD4TB and qXR outperform their predecessors, meeting the standard set in the WHO guideline. Version updates arise rapidly, can involve large changes in the underlying neural network, and are rolled out globally. Independent, evidence-based guidance is urgently needed to help end users prepare for updated technology.

## Supporting information

**S1 Fig. Receiver Operating Characteristic (ROC) Curves of CAD4TB v6 and v7 (left) and qXR v2 and v3 (right).**
(TIF)

**S2 Fig. (A) Sensitivity versus threshold abnormality score for qXRv2 and v3; (B) Xpert tests saved versus threshold abnormality score for qXRv2 and v3; (C) Sensitivity versus threshold abnormality score for CAD4TBv6 and v7; (D) Xpert tests saved versus threshold abnormality score for CAD4TBv6 and v7.**
(TIF)

**S3 Fig. Histograms showing the distribution of abnormality scores of CAD4TB versions 6 and 7 and qXR versions 2 and 3 disaggregated by bacteriological status and by history of TB.**
(TIF)

**S4 Fig. The performance of CAD software versions as Area Under the Receiver Operating Characteristic curve (AUC) stratified by age group, patient source, history of TB, and gender.**
(TIF)

**S1 Table. Comparison of the AUCs of two versions of CAD4TB and qXR.**
(XLSX)

**S2 Table. How threshold score impacts sensitivity and number of diagnostic tests saved for different product versions.**
(XLSX)

**S3 Table. The AUCs of CAD product versions in sub-analyses that showed differences in performance.**
(XLSX)

## Acknowledgments

Delft Imaging Systems and Qure.ai allowed us to use all included CAD products free of charge, but they had no influence on any aspects of our work.

## Author Contributions

**Conceptualization:** Zhi Zhen Qin, Shahriar Ahmed, Mohammad Shahnewaz Sarker, Sayera Banu, Jacob Creswell.

**Data curation:** Zhi Zhen Qin, Shahriar Ahmed, Kishor Paul, Ahammad Shafiq Sikder Adel.

**Formal analysis:** Zhi Zhen Qin, Ahammad Shafiq Sikder Adel.

**Funding acquisition:** Shahriar Ahmed, Kishor Paul, Sayera Banu.

**Investigation:** Zhi Zhen Qin, Sayera Banu, Jacob Creswell.

**Methodology:** Zhi Zhen Qin, Ahammad Shafiq Sikder Adel, Jacob Creswell.

**Project administration:** Sayera Banu, Jacob Creswell.

**Supervision:** Sayera Banu, Jacob Creswell.

**Validation:** Mohammad Shahnewaz Sarker.

**Writing – original draft:** Zhi Zhen Qin, Rachael Barrett.

**Writing – review & editing:** Zhi Zhen Qin, Rachael Barrett, Shahriar Ahmed, Mohammad Shahnewaz Sarker, Kishor Paul, Ahammad Shafiq Sikder Adel, Sayera Banu, Jacob Creswell.

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
