## [Decision Letter · Decision Letter 0]

3 May 2022

PDIG-D-22-00062

Comparing different versions of computer-aided detection products when reading chest X-rays for tuberculosis

PLOS Digital Health

Dear Dr. Qin,

Thank you for submitting your manuscript to PLOS Digital Health. After careful consideration, we feel that it has merit but does not fully meet PLOS Digital Health's publication criteria as it currently stands. Therefore, we invite you to submit a revised version of the manuscript that addresses the points raised during the review process.

Please submit your revised manuscript by . If you will need more time than this to complete your revisions, please reply to this message or contact the journal office at digitalhealth@plos.org. Please include the following items when submitting your revised manuscript:

We look forward to receiving your revised manuscript.

Kind regards,

Gilles Guillot

Academic Editor

PLOS Digital Health

Journal Requirements:

1. Please update your Competing Interests statement. If you have no competing interests to declare, please state: “The authors have declared that no competing interests exist.”

Additional Editor Comments (if provided):

This ms addresses the question of automatic TB detection via AI algorithms applied to Xray chest images. 

This has potentially significant consequences in terms of pathways of care, their efficacy and cost. 

The manuscript lacks a brief intro about the rationale underlying the various AI algorithms compared, their similarity and differences. 

In its current version, the study is purely descritpive about algorithm outcomes. Because algorithms have been described elsewhere, it is possible to 

investigate and discuss at least briefly the sources of difference between algorithms. 

In the overall, this study is useful, has potentially a significant societal impact but is essentially descriptive and provides no insight about the AI processes at stake.

Please consider also reviewers' comments and my specific comments below:

p4l56 Definition of abnormality score not fully clear

p5 l88-89 explain why it makes sense to compare score across two differerent algorithm versions

p6 l108 ref in parentheses unusual and not in ref list

Figures are unreadable and simply can not be assessed. Not clear if the pdf file conversion is the culprit or if original figures were of low quality.

Link to github not formatted properly

Actual github repo MAchineBGD does not seem to exist

 I could access repo BGD_AI_for_TB_Detection and it seems neat

Reviewers' comments:

Reviewer's Responses to Questions

**Comments to the Author**

1. Does this manuscript meet PLOS Digital Health’s publication criteria? Is the manuscript technically sound, and do the data support the conclusions? The manuscript must describe methodologically and ethically rigorous research with conclusions that are appropriately drawn based on the data presented.

Reviewer #1: Partly

Reviewer #2: Yes

2. Has the statistical analysis been performed appropriately and rigorously?

Reviewer #1: I don't know

Reviewer #2: Yes

3. Have the authors made all data underlying the findings in their manuscript fully available (please refer to the Data Availability Statement at the start of the manuscript PDF file)?

Reviewer #1: Yes

Reviewer #2: No

4. Is the manuscript presented in an intelligible fashion and written in standard English?

Reviewer #1: Yes

Reviewer #2: Yes

5. Review Comments to the Author

Reviewer #1: “We therefore compare the performance of two WHO-evaluated CAD 52 product versions with the subsequent versions, using bacteriological evidence as the 53 reference standard. Xpert provided a bacteriological reference standard, 74 confirming the presence (Bac+) or absence (Bac-) of TB.” – could you clarify/confirm if this was for Mycobacterium?

“A dichotomous result (TB-associated 58 abnormalities present or absent) is arrived at by setting a threshold abnormality score, above 59 which the algorithm suggests that TB-associated abnormalities are present, and that the 60 individual should undergo further confirmatory testing.” – Please explain how the threshold abnormality score was set. What technical aspects define a threshold score.

“The image was then read by one of three 69 radiologists registered with the Bangladesh Medical and Dental Council. They classified each image as ‘normal’ or 71 ‘abnormal’ (including any abnormality, whether consistent with TB or not).” – what is the radiologists’ interobserver variability?”

“Both products had been reviewed by the WHO 62 Guidelines Development Group and approved for use in TB triage and screening (in 63 individuals ≥15 years) in 2021.” – Why are patients <15 not included based on WHO recommendations for screening with CAD?

“The human radiologists sensitivity was 88.2% (87.2-89.1%) and specificty was 62.8%, (61.8- 190 63.9%) “ – typo for specificity

“Matching the sensitivity, all versions had significantly greater specificity 191 than human radiologists except for CAD4TB v6, which was similar.” – Please explain technical aspects that attributed to the improvement in specificity with the newer versions. How was improved specificity defined, the ability for the radiologist to diagnose TB versus other forms of pulmonary diseases? If so, what diseases? What diseases were associated with any false positive or false negative interpretations?

Reviewer #2: 1] Overall, the paper is well written and organized. 

2] The tables and figures are well organized. 

3] The paper provides a statistical data analysis of the previous and current WHO recommended CAD products for TB triaging and screening. 

4] The limitations of the study are well stated. 

5] However, the GitHub link provided by the author is not found on the GitHub repository.

6] Neural Network is rather a broad term used in the abstract. Can the authors provide a brief paragraph (after introduction) regarding the previous CAD algorithms implemented in CAD4TB (v6) and qXR (v2) compared to the recent CAD powered by AI CAD4TB (v7) and qXR (v3).

6. PLOS authors have the option to publish the peer review history of their article (what does this mean?). If published, this will include your full peer review and any attached files.

**Do you want your identity to be public for this peer review?** For information about this choice, including consent withdrawal, please see our Privacy Policy.

Reviewer #1: No

Reviewer #2: No

---

## [Editor Report · Decision Letter 1]

15 May 2022

Comparing different versions of computer-aided detection products when reading chest X-rays for tuberculosis

PDIG-D-22-00062R1

Dear Qin,

We are pleased to inform you that your manuscript 'Comparing different versions of computer-aided detection products when reading chest X-rays for tuberculosis' has been provisionally accepted for publication in PLOS Digital Health.

Best regards,

Gilles Guillot

Academic Editor

PLOS Digital Health